# Validation of the AUDIT and AUDIT-C for Hazardous Drinking in Community-Dwelling Older Adults

**DOI:** 10.3390/ijerph18179266

**Published:** 2021-09-02

**Authors:** Yannic van Gils, Erik Franck, Eva Dierckx, Sebastiaan P. J. van Alphen, John B. Saunders, Geert Dom

**Affiliations:** 1Faculty of Medicine and Social Science, Centre for Research and Innovation in Care (CRIC), University of Antwerp, Universiteitsplein 1, 2610 Wilrijk, Belgium; erik.franck@uantwerpen.be (E.F.); Geert.dom@uantwerpen.be (G.D.); 2Faculty of Psychology and Educational Sciences, Vrije Universiteit Brussel, Pleinlaan 2, 1050 Elsene, Belgium; eva.dierckx@vub.be (E.D.); b.van.alphen@mondriaan.eu (S.P.J.v.A.); 3Alexianen Zorggroep Tienen, Psychiatric Hospital, Liefdestraat 10, 3300 Tienen, Belgium; 4Clinical Centre of Excellence for Personality Disorders in Older Adults, Mondriaan Hospital, J.F. Kennedylaan, 301, 6419 XZ Heerlen-Maastricht, The Netherlands; 5Department of Medical and Clinical Psychology, Tilburg University, Warandelaan 2, 5037 AB Tilburg, The Netherlands; 6National Centre for Youth Substance Use Research, The University of Queensland, Brisbane, QLD 4072, Australia; mail@jbsaunders.net; 7Collaborative Antwerp Psychiatric Research Institute (CAPRI), Faculty of Medicine and Social Science, University of Antwerp, Universiteitsplein 1, 2610 Wilrijk, Belgium

**Keywords:** older adults, hazardous drinking, AUDIT, AUDIT-C, validity

## Abstract

Background: One of the best-known tools in screening for hazardous drinking is the Alcohol Use Disorders Identification Test (AUDIT) and its abbreviated form, the AUDIT-C. The aim of the present study is to determine the cut-offs of both instruments in identifying hazardous drinking in older adults. Method: A sample of 1577 older adults completed a questionnaire regarding alcohol behavior. Hazardous drinking was defined as drinking >10 units/week. Receiver operating characteristics (ROC) curves of AUDIT and AUDIT-C were calculated and cut-off scores were derived. Results: Respectively 27.3% and 12.3% of older men and women drank >10 units/week. For the AUDIT the best trade-off between sensitivity and specificity was using a cut-off of ≥5 for men and ≥4 for women, which yielded in men sensitivity and specificity values respectively of 80.7% and 81.3% and in women 100% and 71.7%, respectively. We found the AUDIT-C to perform well with an optimal cut-off of ≥5 for men and ≥4 for women, which generated in men sensitivity and specificity values respectively of 76.5% and 85.3% and in women 100% and 74.1%, respectively. Conclusion: The AUDIT-C is accurate and sufficient in screening for hazardous drinking in community-dwelling older adults if the cut-offs are tailored by gender.

## 1. Introduction

The 65+ segment of our population is rising firmly. Older adults are living longer and can expect to live into their sixties and beyond. By 2050, the number of adults aged 60+ will nearly double from 12% to 22% and is expected to exceed 2 billion. Among the 80+ there will be almost 434 million people worldwide [1]. In addition to the increase in the number of older adults, there is also an increase in alcohol consumption and in parallel unhealthy alcohol use among this population. Grant, B.F., et al. [2] conducted a longitudinal study between 2001 and 2013 and reported an increase of 22% in alcohol use and an increase of 65.2% in high-risk drinking among older adults. In a recent study conducted in four European countries among older adults aged 60–74, 60.3 to 84.7% reported drinking alcoholic beverages [3]. In our previous study among community-dwelling older adults, 38.4% of men and 17.1% of women were categorized as hazardous drinkers (>3 units/day or >7 units/week) [4].

Public mental health emphasizes the importance of health promotion and prevention of mental health problems [5]. Therefore, identifying older adults with hazardous alcohol use is crucial to provide them proper advice and care [6]. There are several distinctive risks associated with hazardous consumption in older people. First, chronic health problems tend to increase with age which increases their odds of developing at least one chronic illness [7]. Alcohol is a toxic drug that can aggravate existing health problems as well as compromise treatment effectiveness and the safety of prescribed drugs [7,8,9]. The interaction between age-related physiological changes and alcohol consumption may cause or intensify serious health problems among older adults [9]. Thus, alcohol consumption is a risk factor for many chronic diseases and conditions. Moreover, studies encounter the significance of alcohol use to substantial health loss [10,11].

Second, previous studies have frequently reported a relationship between alcohol consumption and injuries among older adults. In a study of Leclerc, B.S., et al. [12], older adults who drank alcohol on a regular basis were at high risk of becoming recurrent fallers. Also, according to Pluijm, S.M.F., et al. [13], higher levels of drinking (≥18 drinks per week) are an important predictor in the risk profile for recurrent falling in a sample of Dutch community dwelling older adults. Even moderate alcohol use (≤2 drinks per day) was associated with increased falls among older adults [14]. Alcohol use can negatively influence one’s motor skills which may lead to the assumption that age-related health risks like falling might increase with the level of alcohol consumption [7]. Recently, different countries and regions (e.g., Flanders in 2016 https://www.vad.be/assets/richtlijn-voor-alcoholgebruik, (accessed on 19 February 2021) and Australia in 2020 https://www.health.gov.au/news/australian-alcohol-guidelines-revised, (accessed on 19 February 2021) have revised their “healthy” drinking guidelines, proposing a maximum alcohol consumption of 10 units a week, independent of age. Additionally, this guideline has been recently described as ‘lowest risk of all-cause mortality’ in a study among current drinkers [15].

A third risk lies in taking medication and alcohol at the same time. Older people are the largest per capita consumers of prescription medication [7,16]. In Pringle, K.E., et al. [17] 77% of older adults using medication reported taking at least one alcohol-interactive medication. There are many hazards associated with medication used by older adults and one of them is how alcohol interacts with prescription medication. This might be explained by age-related changes in the absorption, distribution, and metabolism of alcohol and medicine which may alter the effects of medication [7,18].

Screening for hazardous drinking patterns can foster prevention [19]. However, several studies indicate that alcohol problems among older adults are frequently undetected or misdiagnosed, which can prolong or worsen health problems associated with hazardous alcohol use [20,21]. A tailored screening tool for older adults may intercept this problem. Various screening tests have been developed to detect hazardous drinking. One of the best-known and validated tools is the Alcohol Use Disorder Identification Test (AUDIT), which is used worldwide [22]. The AUDIT was developed by the World Health Organization (WHO) for screening in primary health care. It consists of 10 items and identifies (1) the frequency and quantity of alcohol use and hazardous use, (2) consumption that has caused harm, and (3) possible dependence on alcohol [23], resulting in a total score of 40. An abbreviated version has also been developed, the AUDIT-C (AUDIT-Consumption) which involves the first three questions on quantity and frequency of alcohol use [24], resulting in a total score of 12. Studies are inconsistent about the use of the full version of the AUDIT in older populations. AUDIT was evaluated by [19] among 242 Korean men aged 65 and reported a good sensitivity and specificity in identifying at-risk drinking (>7 units/week) with an AUDIT score ≥7. Aalto, M., et al. [25] used a random general sample of 517 older adults aged 65–74 and suggested to lower the cut-off score to ≥5 for both older men and women to optimize the performance of the AUDIT to detect hazardous drinking (≥8 units/week or ≥4 units/day). On the other hand, Källmén, H., et al. [26] evaluated the reliability and validity of the AUDIT in 1066 Swedish adults aged 70–80 years and compared the results with a general population. They stated that the performance of the AUDIT was less satisfactory in this older population in comparison to a younger one, which might cast doubt on the appropriateness of the AUDIT for the screening of hazardous drinking in older adults. Considering the AUDIT-C, Dreher-Weber, M., et al. [27] conducted a study on the validity of this tool among 344 nursing home residents. Their data suggested an AUDIT-C cut-off of ≥4 for men and ≥2 for women to detect a consumption of ≥10 g of alcohol per day. Aalto, M., et al. [25] proposed an AUDIT-C cut-off of ≥4 for both older men and women. Recently, Stewart, D. et al. [28] conducted a study in the UK with 143 older adults recruited in non-clinical settings. They examined the validity of the AUDIT-C and reported a good performance in identifying unhealthy alcohol use (>14 units/week) in older adults when using a cut-off of ≥5 for both older men and women.

Because of the increase of alcohol use in this fast-growing segment of our population and because of the inconsistencies in previous research (mostly due to methodological issues), the relatively small sample sizes and the overall paucity of research among older adults, the main objective of this study is to evaluate the validity of the AUDIT and AUDIT-C and optimal cut-offs in detecting hazardous drinking within a large sample of community-dwelling older adults.

## 2. Materials and Methods

This is a study exploring the sensitivity and specificity of the AUDIT and AUDIT-C in detecting hazardous alcohol use in community-dwelling older men and women.

### 2.1. Sample

This study is part of a larger research project on the drinking patterns of community-dwelling older adults in Belgium. The inclusion criteria were being 60+, living at home, and having a good comprehension of the Dutch language. The latter was essential because the questionnaires were only available in Dutch. Older adults reporting memory problems, having a neurodegenerative disease, or with sensory deficits were excluded. The sample population, registered from October 2013 to April 2019, involved 1971 older adults living in the Flemish part of Belgium (Flanders). Non-drinkers were defined as those who indicated never consuming alcohol and were removed (*n* = 394). The final drinking sample included 1577 respondents.

### 2.2. Process of Data Collection

A snowball sampling was used to recruit our participants. The objective and process of the study were presented during gatherings in community centers and local activity groups. Interested individuals could indicate their willingness to participate in the study. At that point, they were asked if they wanted to complete the questionnaire at the time or if they wanted to complete it at home. If the participant wanted to complete the questionnaire at home, they were given an appointment so that one of the researcher assistants could come by with the questionnaire. The researcher assistant was present should the participant have any questions or concerns. When a couple was willing to participate, both spouses were spaced far enough apart from each other to reduce potential influence on each other. After they finished filling out the questionnaire, they were asked if they knew of any people who would be willing to participate in the study as well.

### 2.3. Statements of Ethical Approval

The research protocol was approved by the Ethical Committee of University Hospital in Antwerp (reference 14/44/458). Anonymity and confidentiality were emphasized by the interviewer. A written informed consent was obtained before starting the survey: no names were registered and all the obtained data were processed by the research team.

### 2.4. Measurements

The Alcohol Use Disorder Identification Test (AUDIT) [23,24] was used to assess hazardous alcohol use. It includes ten questions: three involving quantity and frequency of alcohol use, three about alcohol dependence, and four relating to problems caused by alcohol misuse. Items are scored between 0 to 4, which implies a total score range of 0–40. Cronbach’s alpha for the full scale (AUDIT) in our drinking sample was 0.72. Using the guideline of ≥0.70 makes this value at the limit of acceptable for internal consistency [29]. The full AUDIT was included, and from it the abbreviated version (AUDIT-C) was derived and scored in a standard manner [24]. For the AUDIT-C, the average inter-item correlation (AIC) was calculated, as this measure of internal consistency is independent of the number of items in a scale (as opposed to Cronbach’s alpha). The AIC of the scales was 0.33, which is acceptable using the range between 0.15 and 0.50 as rule of thumb [30].

The recommended Belgian guidelines level of >10 units/week were applied as a gold standard. To calculate the total weekly alcohol consumption, the first two questions of the AUDIT were used. Respondents were given the full version of the AUDIT questionnaire with the original questions. In order to calculate the gold standard, we used the original first two questions of the AUDIT, but participants were given a more elaborate set of answers. The adapted answer segments were the following: for the first question ‘How often do you have a drink containing alcohol?’ we used the following possibilities: never, monthly or less, 1 per week, 2 per week, 3 per week, 4 per week, 5 per week, 6 per week, every day. The second question ‘How many drinks containing alcohol do you have on a typical day when you are drinking?’ became an open question. This allowed the respondents to specify any quantity and they were not bound by the predefined categories proposed in the original version of the AUDIT. To obtain the original scores of the AUDIT and AUDIT-C, we recoded these answers to the original version (Appendix A).

### 2.5. Statistical Analyses

All statistical analyses were conducted using SPSS (IBM Corp. Released 2017. IBM SPSS Statistics for Windows, Version 26.0. Armonk, NY: IBM Corp.).

First, prevalence of hazardous drinking and mean AUDIT and AUDIT-C scores were described for the total drinking sample and by gender.

In order to operationalize ‘hazardous drinking’, the reference category was defined according to the current Flemish safe drinking guidelines as drinking 10 or more units per week (1 unit is 10 g of ethanol) for both men and women (www.vad.be; accessed on 19 February 2021). Subsequently, non-hazardous drinking was defined as drinking less than 10 units per week.

To represent the accuracy of the test we determined the area under the curve (AUC) using Receiver Operating Characteristics (ROC) curve analysis [31]. AUC ≥0.80 is generally considered adequate [32]. The AUC was measured for the total drinking sample as well as for the gender subgroups. Also, the Youden function was calculated to determine the optimal cut-offs for the AUDIT and AUDIT-C for men and women separately [33].

Next, sensitivity, specificity, positive predicted value (PPV) and negative predicted value (NPV) were calculated for each AUDIT and AUDIT-C cut-off score for the total drinking sample and for the male/female subgroups.

## 3. Results

### 3.1. Sample

Table 1 represents the description of demographic variables and drinking behavior of the total drinking sample and the gender subsamples. A total of 308 older adults (19.5%) were identified as hazardous drinkers based on the drinking guidelines. There was a higher prevalence of hazardous drinking among men (27.3%) than women (12.3%), ꭕ2 (1) = 56.52; *p* < 0.001. The mean scores on the AUDIT and AUDIT-C were respectively M = 3.89 (SD = 2.99) and M = 3.44 (SD = 1.92). Compared to women, men had higher mean AUDIT (men: M = 4.65, SD = 3.51–women: M = 3.19, SD = 2.19; t(1569) = −9.958; *p* < 0.001, d = 0.48) and AUDIT-C scores (men: M = 4.01, SD = 2.09–women: M = 2.91, SD = 1.56; t(1572) = −11.889; *p* < 0.001, d = 0.57). This section may be divided by subheadings. It should provide a concise and precise description of the experimental results, their interpretation, as well as the experimental conclusions that can be drawn.

### 3.2. ROC Analysis: Validity of Screening Tools AUDIT and AUDIT-C for Hazardous Drinking

ROC curve analysis, using the recommended Belgian guidelines level of >10 units/week, showed significant results for the AUDIT and AUDIT-C for the total sample as well as for both gender subgroups (Table 2). For the AUDIT, the AUC for the total sample is 0.905 (95% C.I. = 0.890–0.921, *p* < 0.001), 0.903 for men (95% C.I. = 0.881–0.925, *p* < 0.001) and 0.897 for women (95% C.I. = 0.873–0.920, *p* < 0.001), respectively. For the AUDIT-C, the AUC for the total sample is 0.920 (95% C.I. = 0.906–0.934, *p* < 0.001), with an AUC for men of 0.920 (95% C.I. = 0.899–0.941, *p* < 0.001) and an AUC for women of 0.909 (95% C.I. = 0.887–0.930, *p* < 0.001).

The sensitivities and specificities for different cut-offs are shown in Table 3. For the total sample, the optimal cut-off score for the AUDIT in screening for hazardous drinking, as determined by the Youden function in the ROC analysis, was 5, which resulted in a sensitivity of 73.5%, specificity of 86.1%, PPV of 56%, and NPV of 93.4%. For men, the optimal cut-off score for the AUDIT in screening for hazardous drinking was 5, with sensitivity of 80.7%, specificity of 81.3%, PPV of 60.6%, and NPV of 92.2%. For women, the optimal cut-off score for the AUDIT in screening for hazardous drinking was 4, with sensitivity of 100%, specificity of 71.7%, PPV of 32.2%, and NPV of 100%. For the AUDIT-C the optimal cut-off for the total sample is 5, with sensitivity of 69.2%, specificity of 90.8%, PPV of 63.5%, and NPV of 92.7%. For the AUDIT-C among men, the optimal cut-off for the total sample is 5, with sensitivity of 76.5%, specificity of 85.3%, PPV of 65%, and NPV of 91.1%. For the AUDIT-C the optimal cut-off for women is 4, with sensitivity of 100%, specificity of 74.1%, PPV of 34.1%, and NPV of 100%.

## 4. Discussion

The present study used the AUDIT and AUDIT-C as screening tools for hazardous drinking and investigated each tool’s AUC and optimal cut-off scores.

### 4.1. The AUDIT

The ROC analysis for the AUDIT showed that 89% of men and 90% of women could be classified correctly in terms of hazardous alcohol use using the AUDIT. The best trade-off between sensitivity and specificity was using a cut-off of ≥5 for men and ≥4 for women, which yielded in men sensitivity and specificity values of 80.7% and 81.3%, respectively, and in women 100% and 71.7%, respectively. Our cut-off scores are partially in line with previous studies. Aalto, M., et al. [25] suggested a cut-off score of ≥5 (sensitivity: 0.86 and specificity: 0.87) to detect heavy drinking (≥8 units/week or ≥4 units/day), however for both older men and women. Ryou, Y.I., et al. [19] proposed a cut-off score of ≥7 (sensitivity: 0.77 and specificity: 0.85) for older men to detect the consumption of >7 units/week, which is much higher than ours. Despite the fact that both studies were conducted among older adults, they did not deal with the possibility of gender differences in their analyses and used relatively smaller samples. This makes comparisons between the results difficult.

### 4.2. The AUDIT-C

The corresponding ROC analyses for the AUDIT-C indicated that 92% of men and 90% of women could be classified correctly using the AUDIT. We found the AUDIT-C to perform well in identifying hazardous drinking, with an optimal cut-off of ≥5 for men and ≥4 for women, which yielded in men sensitivity and specificity values of 76.5% and 85.3%, respectively, and in women 100% and 74.1%, respectively. Our results are not quite in line with previous studies: Dreher-Weber, M., et al. [27] suggested an AUDIT-C cut-off of ≥4 (sensitivity: 0.70 and specificity: 0.83) for older men and ≥2 (sensitivity: 0.70 and specificity: 0.83) for older women to detect the consumption of ≥10 g of alcohol per day, which is a lower threshold than we adopted. These cut-offs are stricter, but are intended for nursing home residents. Older adults in nursing home might be more fragile and vulnerable to the adverse consequence of alcohol which might explain the lower threshold for hazardous alcohol use and the lower cut-offs. Aalto, M., et al. [25] proposed an AUDIT-C cut-off of ≥4 (sensitivity: 0.94 and specificity: 0.80) for both older men and women but did not conduct their analyses separately in men and women. They acknowledge the possibility that the AUDIT questionnaire might perform differently in men and women. Finally, Stewart, D. [28] recommended a cut-off of ≥5 (sensitivity: 0.88 and specificity: 0.85) in both older men and women to detect the consumption of ≥14 units/week. They also did not analyze the validity of the questionnaire separately for men and women, which makes comparison challenging.

### 4.3. Implications

#### 4.3.1. The AUDIT vs. the AUDIT-C

Interestingly, our results suggested the same cut-offs for the AUDIT and AUDIT-C. This might be a starting point for focusing on quantity and frequency of drinking while screening for hazardous drinking in older adults. The full version of the AUDIT consists of items about alcohol dependence and alcohol-related problems. These items might not be applicable for older adults probably because they are uncommon in this segment of the population. Based upon our findings and due to practical considerations, we propose to use the AUDIT-C to assess hazardous drinking in older adults. Indeed, the AUDIT-C is shorter (three items instead of ten), more suitable, and easier to use, which might be more appropriate in routine check-ups by general practitioners as well as in research. The full AUDIT can be reserved for populations where the experience of existing alcohol-related harm and/or dependence is likely to be higher or is suspected. In addition, further research could verify in which way the items that are not part of the AUDIT-C are associated with hazardous drinking.

#### 4.3.2. Men vs. Women

We propose a different cut-off score for men and women. By maximizing the Youden’s Index [33] we were able to find, from the ROC-curve, an optimal cut-off point independently of the prevalence. According to this index, a stricter approach for women might be more apt. This is in line with the assumption that women may be more vulnerable to the adverse consequences of alcohol. Physiologically, women have less lean body mass than men which makes them less tolerant to alcohol [34,35]. Furthermore, older women tend to have higher rates of polypharmacy than older men [36] which makes them more susceptible for the possible interactions between alcohol and prescribed medication [34].

### 4.4. Strengths & Limitations

The strength of this study lies in the large and representative sample of community-dwelling older adults. To our knowledge, previous researchers conducted their studies in much smaller samples, which may limit generalization of their results. Because of our large sample, our results may be more suitable for use in clinical practice as well as in further research. Additional research could verify how well the cut-off scores will work in independent samples.

One concern is the snowball sampling as it may limit the representativeness of our sample versus the total population of older adults. It might be that participants will associate with similar types of drinkers to themselves. Also, the educational level of our sample was quite high. This might be explained by the snowball bias as higher educated older adults will have the tendency to socialize with peers. This may limit the wider applicability of the findings, as does the lack of information about people with lower levels of education and those who did not volunteer to participate. However, snowball sampling might be particularly beneficial when a study which involves private matters, such as alcohol use might be for older adults [37]. Secondly, it is well known that some people underestimate their alcohol intake for various reasons. This tendency might be a result of cognitive impairment [27] but may also be more likely in older adults in whom hazardous drinking might have a negative connotation [38]. Higher levels of social desirability have been reported when a situation has been judged as more unethical [39]. Alcohol misuse might be seen as unethical by the ‘older’ older adults of our sample. Additionally, women and individuals with higher levels of religiousness have the tendency to report higher levels of social desirability [39]. Both are represented in our sample which may lead to under-reporting alcohol use. The latter will affect the performance of screening tools that contain quantity and frequency items, such as the AUDIT and AUDIT-C [38]. Further research could focus on a population with heavy alcohol use and misuse to investigate if these cut-off scores are also applicable to them. A third concern might be our lower levels of PPV in both the AUDIT and AUDIT-C. This can be explained by the association between a lower prevalence (in this case of heavy alcohol use) and a poor PPV [40]. Further research could focus on conducting analyses in a population with higher levels of heavy alcohol use in order to counter this issue.

## 5. Conclusions

According to this study, screening for hazardous alcohol use in older adults should focus more on consumption patterns than on alcohol related diagnoses, which are often not adjusted to older adults [27]. In this study, the AUDIT-C, with a cut-off for men ≥5 of and for women of ≥4, showed good ROC values for detecting hazardous drinking in men and women aged 65 and older. The full AUDIT could be reserved for populations where the prevalence of alcohol-related harm is likely to be high or for second-phase assessment for those with AUDIT-C scores above the cut-off points.

## Figures and Tables

**Table 1 ijerph-18-09266-t001:** Descriptive of the total drinking sample (*n* = 1.577) and the gender subsamples.

		Total Drinking Sample (*n =* 1.577)		Men (*n* = 759)		Women (*n* = 818)	
DEMOGRAPHIC VARIABLES							
Age M (SD)		71.97	(6.68)	72.2	(6.71)	71.74	(6.66)
Educational level							
	lower secondary *n* (%)	472	(30)	196	(25.8)	276	(33.9)
	higer secondary *n* (%)	558	(35.5)	255	(33.6)	303	(37.2)
	higher education *n* (%)	543	(34.5)	308	(40.6)	235	(28.9)
Living situation							
	living with a partner *n* (%)	1176	(74.6)	638	(84)	538	(65.9)
	in a relationship without living together *n* (%)	40	(2.5)	23	(3)	17	(2.1)
	divorced/single *n* (%)	139	(8.9)	49	(6.6)	88	(10.7)
	widow *n* (%)	218	(13.8)	47	(6.4)	171	(21)
DRINKING BEHAVIOR							
Drinking frequency	less than weekly *n* (%)	320	(20.3)	110	(14.5)	209	(25.6)
	1/week *n* (%)	363	(23)	159	(20.9)	204	(25)
	2/week *n* (%)	217	(13.8)	107	(14.1)	110	(13.5)
	3/week *n* (%)	164	(10.4)	89	(11.7)	75	(9.2)
	4/week *n* (%)	105	(6.7)	56	(7.4)	49	(6)
	5/week *n* (%)	64	(4.1)	39	(5.1)	25	(3.1)
	6/week *n* (%)	42	(2.7)	25	(3.3)	17	(2.1)
	every day *n* (%)	302	(19.2)	174	(22.9)	127	(15.6)
Drinking >10 units/week * *n* (%)		308	(19.5)	207	(27.3)	100	(12.3)
Drinking quantity on a typical day M (SD)		2.1	(1.4)	2.48	(1.7)	1.75	(0.9)
AUDIT score M (SD) **		3.89	(3.0)	4.65	(3.5)	3.19	(2.2)
AUDIT-C score M (SD) ***		3.44	(1.9)	4.01	(2.1)	2.91	(1.6)

M Mean; SD Standard Deviation; * Hazardous drinking according to the Belgian VAD guidelines >10 units/week; ** Range 0–40; *** Range 0–12.

**Table 2 ijerph-18-09266-t002:** Area Under the Curve (AUC).

	AUDIT				AUDIT-C			
	AUC	95% C.I.		*p*-Value	AUC	95% C.I.		*p*-Value
		Lower Bound	Upper Bound			Lower Bound	Upper Bound	
Total sample	0.905	0.890	0.921	<0.001	0.920	0.906	0.934	<0.001
Men	0.898	0.873	0.920	<0.001	0.920	0.899	0.941	<0.001
Women	0.903	0.881	0.925	<0.001	0.909	0.887	0.930	<0.001

**Table 3 ijerph-18-09266-t003:** Sensitivity, specificity, PPV, and NPV in percentages for different cut-off scores AUDIT and AUDIT-C for the total sample and the male/female subgroups.

	Cut-Off Score AUDIT				Cut-Off Scores AUDIT-C			
	3	4	5	6	3	4	5	6
Total Sample								
Sensitivity	100	100	73.5	54.8	100	100	69.2	48.4
Specificity	43.4	65.1	86.6	93.3	45.0	67.8	90.8	97.1
PPV	29.06	39.91	56.01	65.66	29.61	41.82	63.48	79.41
NPV	100	100	93.36	89.91	100	100	92.71	89.04
Women								
Sensitivity	100	100	58.2	31.9	100	100	53.8	28.6
Specificity	50.6	71.7	90.7	95.1	52.0	74.1	95.0	99.0
PPV	21.36	32.15	45.68	46.77	21.82	34.08	59.03	78.78
NPV	100	100	94.18	91.23	100	100	93.88	91.18
Men								
Sensitivity	100	100	0.807	66.3	100	100	76.5	58.3
Specificity	34.0	56.5	81.3	91.0	35.8	59.6	85.3	94.7
PPV	35.08	45.06	60.64	72.51	35.68	46.86	65	79.56
NPV	100	100	92.20	88.33	100	100	91.05	86.43

## Data Availability

The data is available upon request from the corresponding authors.

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
