# Peer review of "Validation of the AUDIT and AUDIT-C for Hazardous Drinking in Community-Dwelling Older Adults"

_ijerph, 2021, doi:10.3390/ijerph18179266_

Round 1

Reviewer 1 Report

It is valuable work to validate AUDIT and AUDIT-C for the elderly and establish standards or norms for the cutoff score. If authors understand the limitations of the study, interpret the results and conclude, and supplement a few things, it will be sufficient to be published in this journal. The things I think should be revised and supplemented are as follows.

  1. Please divide the second paragraph of the introduction into several paragraphs for readability.
  2. Because producing the cutoff score is one of the standardization process, the representation of the sample is very important. Please be more specific about the representation of the sample. Please describe in more detail the representation of the sample. Of course, No matter how well-organized sample cannot represent total population, please describe the limitation of sample as well. Although you point out that snowball sampling is a kind of limitation of study, it may not be appropriate for this kind of validation or standardization research because it is highly likely that participants will introduce similar types of drinkers to themselves. Even so, you have to say that it is a significant result.
  3. Although AUDIT is a widely used scale worldwide, participants may respond in a socially desirable way because it is a self-reported measure and deals with alcohol abuse or addiction. It is better to discuss that this may vary by demographic variable such as gender or culture.
  4. Even if IRB approval and informed consent are included in the latter section, you should explain the process of data collection considering the ethical aspects of research in research method section.
  5. I know that this journal regulation does not insert the year when presenting the author’s name while presenting the quote.

Author Response

Reviewer 1

It is valuable work to validate AUDIT and AUDIT-C for the elderly and establish standards or norms for the cutoff score. If authors understand the limitations of the study, interpret the results and conclude, and supplement a few things, it will be sufficient to be published in this journal. The things I think should be revised and supplemented are as follows.

We thank the reviewer for their appreciation of our manuscript and for the valuable suggestions. In line with the suggestions, we have changed the manuscript. The changes are highlighted in the text.

Please divide the second paragraph of the introduction into several paragraphs for readability.

We have changed the text accordingly. We divided the second paragraph into four sections.

Because producing the cutoff score is one of the standardization process, the representation of the sample is very important. Please be more specific about the representation of the sample. Please describe in more detail the representation of the sample.

We added in table 1 demographic information:

Table 1 represents the descriptive of demographic variables and drinking behavior of the total drinking sample and the gender subsamples.

Table 1. Descriptive of the total drinking sample (n 1.577) and the gender subsamples

Of course, No matter how well-organized sample cannot represent total population, please describe the limitation of sample as well. Although you point out that snowball sampling is a kind of limitation of study, it may not be appropriate for this kind of validation or standardization research because it is highly likely that participants will introduce similar types of drinkers to themselves. Even so, you have to say that it is a significant result.

We have changed the text accordingly:

One concern is the snowball sampling as it may limit the representativeness of our sample versus the total population of older adults. It might be that participants will associate with similar types of drinkers to themselves. Also, the educational level of our sample was quite high. This might be explained by the snowball bias as higher educated older adults will have the tendency to socialise with peers. This may limit the wider applicability of the findings, as does the lack of information about people with lower levels of education and those who did not volunteer to participate.

Although AUDIT is a widely used scale worldwide, participants may respond in a socially desirable way because it is a self-reported measure and deals with alcohol abuse or addiction. It is better to discuss that this may vary by demographic variable such as gender or culture.

We have changed the text accordingly:

This tendency might be a result of cognitive impairment [27] but may also be more likely in older adults in whom hazardous drinking might have a negative connotation [38]. Higher levels of social desirability has been reported when a situation has been judged as more unethical [39]. Alcohol misuse might be seen as unethical by the ‘older’ older adults of our sample. Additionally, women and individuals with higher levels of religiousness have the tendency to report higher levels of social desirability [39]. Both are represented in our sample which may lead to under-reporting alcohol use. The latter will affect the performance of screening tools that contain quantity and frequency items, such as the AUDIT and AUDIT-C [38].

Even if IRB approval and informed consent are included in the latter section, you should explain the process of data collection considering the ethical aspects of research in research method section.

We have changed the text accordingly.

  1. we changed in the research method section ‘Procedure’ into ‘Process of data collection’
  2. we added the IRB code in the segment ‘statement of ethical approval’

The research protocol was approved by the Ethical Committee of University Hospital in Antwerp (reference 14/44/458). Anonymity and confidentiality were emphasized by the interviewer. A written informed consent was obtained before starting the survey: no names were registered and all the obtained data were processed by the research team.

I know that this journal regulation does not insert the year when presenting the author’s name while presenting the quote.

Thank you, we have changed the text accordingly.

Reviewer 2 Report

The is a study that used a community based sample of older adults to assess the validity of the AUDIT and short-form AUDIT-C to diagnose hazardous drinking. Using a sample of over 1500 participants dwelling in Flanders Belgium participants completed a questionnaire that included the AUDIT. Hazardous drinking was defined using by the Flemish safe drinking guidelines, which is 10 or more units per week. If I read correctly the first two questions from the AUDIT were used to calculate drinking and define the outcome variable.

The authors found both 27% of the men and 12% of the women met the threshold for hazardous drinking. The AUDIT and AUDIT-C had good classification rate for both men and women separately with areas under the curve from a ROC analysis of about 0.9. The optimal cut-off for diagnosing hazardous drinking was found to be 5 for men and 4 for women, for both the AUDIT and AUDIT-C. The authors recommend the use of the AUDIT-C for screening purposes.   

I thought this was an interesting study with important relevance to a growing at-risk population. The introduction was written well and sets up nicely the rationale for the study. I do have some comments regarding the design and analysis, which are listed below. I think if the authors address the concerns it would strengthen the manuscript.

  1. It is not clear what exactly is contained in the questionnaire administered to participants. Was it just the AUDIT or were there additional questions? If additional questions a brief description of those questions would be helpful.

  1. The reliability of the AUDIT was reported (alpha=.72) but not for the AUDIT-C. Because the AUDIT-C was used in the analyses it would be good to also report the reliability for that measure.

  1. The operational definition for hazardous drinking appeared to be derived from the scale used to predict hazardous drinking. As written, it reads as if the first two items of the AUDIT were used to derive hazardous drinking. And then the AUIDT is assessed in how well it predicts hazardous drinking. If correct, this is a major limitation and needs to be acknowledged. A better design would have been for the outcome to be derived independent of the predictor being assessed. If I misinterpreted the derivation of hazardous drinking, then at least there needs to be clarification in the methods section so other readers are not also confused.

  1. The same cut-score for the AUDIT and AUDIT-C is informative and suggestive that the items not part of the AUDIT-C are not contributing to predicting hazardous drinking. It would be worth pursuing further by conducting a logistic regression where each item serves as a predictor and hazardous drinking as the outcome to verify the later items are not associated with hazardous drinking.

  1. When investigating the optimal cut-score for classification it is recommended, when the sample is large enough, to divide the sample into a training set and a testing set. The cut-score is found using the training set and then that cut-score is used on the testing set to verify how well it works on an independent sample. Only using a testing set can capitalize on chance and the correct classification rates may not carry over and be as high in other samples.   

Author Response

Reviewer 2

The is a study that used a community based sample of older adults to assess the validity of the AUDIT and short-form AUDIT-C to diagnose hazardous drinking. Using a sample of over 1500 participants dwelling in Flanders Belgium participants completed a questionnaire that included the AUDIT. Hazardous drinking was defined using by the Flemish safe drinking guidelines, which is 10 or more units per week. If I read correctly the first two questions from the AUDIT were used to calculate drinking and define the outcome variable.

The authors found both 27% of the men and 12% of the women met the threshold for hazardous drinking. The AUDIT and AUDIT-C had good classification rate for both men and women separately with areas under the curve from a ROC analysis of about 0.9. The optimal cut-off for diagnosing hazardous drinking was found to be 5 for men and 4 for women, for both the AUDIT and AUDIT-C. The authors recommend the use of the AUDIT-C for screening purposes.  

I thought this was an interesting study with important relevance to a growing at-risk population. The introduction was written well and sets up nicely the rationale for the study. I do have some comments regarding the design and analysis, which are listed below. I think if the authors address the concerns it would strengthen the manuscript.

We thank the reviewer for their appreciation of our manuscript and for the valuable suggestions. In line with the suggestions, we have changed the manuscript. The changes are highlighted in the text.

It is not clear what exactly is contained in the questionnaire administered to participants. Was it just the AUDIT or were there additional questions? If additional questions a brief description of those questions would be helpful.

 We have changed the text accordingly:

The recommended Belgian guidelines level of >10units/week were applied as gold standard. To calculate the total weekly alcohol consumption, the first two question of the AUDIT were used. Respondents were given the full version of the AUDIT questionnaire with the original questions. In order to calculate the gold standard, we used the original first two questions of the AUDIT, but participants were given a more elaborate set of answers. The adapted answer segments were the following: for the first question ‘How often do you have a drinking containing alcohol?’ we used the following possibilities: never, monthly of less, 1 per week, 2 per week, 3 per week, 4 per week, 5 per week, 6 per week, every day. The second question ‘How many drinks containing alcohol do you have on a typical day when you are drinking?’ became an open question. This allowed the respondents to specify any quantity and they were not bound by the predefined categories proposed in the original version of the AUDIT. To obtain the original scores of the AUDIT and AUDIT-C, we recoded these answers to the original version (Appendix A).

Appendix A

To obtain the original scores of the AUDIT and AUDIT-C, we recoded these answers to the original version (Table 4).

Table 4. The original and altered answer segments of question 1 and question 2 of the AUDIT

The reliability of the AUDIT was reported (alpha=.72) but not for the AUDIT-C. Because the AUDIT-C was used in the analyses it would be good to also report the reliability for that measure.

We doubt whether the Cronbach alpha is a valid measure when the scale comprises only 3 items. Cronbach's alpha is an old measure of scale reliability and there are several assumptions and constraints on its use.  Originally 8+ items were considered necessary for a reliable scale, however, recently it has been believed that for most purposes 4-6 items are sufficient (Samuels, 2015). Short scales reduce the risk of Cronbach’s alpha inflation and misinterpretation. Calculating alpha for the full AUDIT is more reasonable, though the AUDIT captures hazardous alcohol use (Q1-3), alcohol dependence (Q4-6) and alcohol related harm/harmful alcohol use (Q7-10).  These are different constructs conceptually, strongly correlated with each other, but not expected to be part of a unidimensional scale.

Samuels, P. (2015). Advice on Reliability Analysis with Small Samples. ResearchGate. DOI:10.13140/RG.2.1.1495.5364

We have changed the text accordingly:

The Alcohol Use Disorder Identification Test (AUDIT) [23, 24] was used to assess hazardous alcohol use. It includes ten questions: three involving quantity and frequency of alcohol use, three about alcohol dependence, and four relate to problems caused by alcohol misuse. Items are scored between 0 to 4, which implies a total score range of 0-40. Cronbach’s alpha for the full scale (AUDIT) in our drinking sample was .72. Using the guideline of ≥.70 makes this value at the limit of acceptable for internal consistency [29]. The full AUDIT was included, and from it, the abbreviated version (AUDIT-C) was derived and scored in a standard manner [24]. For the AUDIT-C, the average inter-item correlation (AIC) was calculated, as this measure of internal consistency is independent of the number of items in a scale (as opposed to Cronbach’s alpha). The AIC of the scales was .33, which is acceptable using the range between .15 and .50 as rule of thumb [30].

The operational definition for hazardous drinking appeared to be derived from the scale used to predict hazardous drinking. As written, it reads as if the first two items of the AUDIT were used to derive hazardous drinking. And then the AUIDT is assessed in how well it predicts hazardous drinking. If correct, this is a major limitation and needs to be acknowledged. A better design would have been for the outcome to be derived independent of the predictor being assessed. If I misinterpreted the derivation of hazardous drinking, then at least there needs to be clarification in the methods section so other readers are not also confused.

The outcome (>10units/week) has not been derived by original items of the predictor (AUDIT).

The first two original questions were used but we adapted the answering segment in order to gather more elaborated information concerning the quantity and frequency of alcohol use. So, we used the original questions but we did not used the original answering segment. We altered the answering segment to be able to have an accurate total weekly consumption of our respondents. The original answering segment will not have given use the precise total weekly consumption. Therefore, we needed to adapt the original answering segment.

When the total scores of the AUDIT and AUDIT-C were used as predictors, the answers of the first two questions needed to be recoded into the original version.

The same cut-score for the AUDIT and AUDIT-C is informative and suggestive that the items not part of the AUDIT-C are not contributing to predicting hazardous drinking. It would be worth pursuing further by conducting a logistic regression where each item serves as a predictor and hazardous drinking as the outcome to verify the later items are not associated with hazardous drinking.

This is a proposal well worth checking out.

We have changed the text accordingly:

The full AUDIT can be reserved for populations where the experience of existing alcohol-related harm and/or dependence is likely to be higher or is suspected. In addition, further research could verify in which way the items that are not part of the AUDIT-C are associated with hazardous drinking.

When investigating the optimal cut-score for classification it is recommended, when the sample is large enough, to divide the sample into a training set and a testing set. The cut-score is found using the training set and then that cut-score is used on the testing set to verify how well it works on an independent sample. Only using a testing set can capitalize on chance and the correct classification rates may not carry over and be as high in other samples.  

We took the feedback into consideration and made it into a proposal for further research.

We have changed the text accordingly:

The strength of this study lies in the large and representative sample of community-dwelling older adults. To our knowledge, previous researchers conducted their studies in much smaller samples, which may limit generalization of their results. Because of our large sample, our results may be more fitted to be used in clinical practice as well as in further research. Additional research could verify how well the cut-off scores will work in independent samples. 
